# Investigation of the Interaction between Hearing Function and Comorbidities in Adults Living with Human Immunodeficiency Virus

**DOI:** 10.3390/ijerph182212177

**Published:** 2021-11-19

**Authors:** Ben Sebothoma, Katijah Khoza-Shangase

**Affiliations:** Department of Speech Pathology and Audiology, University of the Witwatersrand, Johannesburg 2050, South Africa; Katijah.Khoza-Shangase@wits.ac.za

**Keywords:** adults, comorbidities, hearing function, HIV, South Africa

## Abstract

Adults living with the human immunodeficiency virus (HIV) have a high prevalence of co-existing comorbidities. While research indicates that adults living with HIV are at risk of developing hearing impairment, limited research exists on the interaction between hearing function and comorbidities in this population. The objective of this study was to determine and compare the hearing function of a group of adults living with HIV and comorbidities and those without comorbidities. A sample of 132 adults living with HIV underwent a basic audiological test battery to assess their hearing function. Participants with comorbidities were 1.23 times more likely to develop hearing loss, with crude odds of 1.236 (95%CI 0.5467 to 2.795), while those with three comorbidities were 2.52 times more likely to develop hearing loss. Participants with hypertension were 93% more likely to develop hearing loss when compared to nonhypertensive participants (OR = 1.928; 95%CI: 0.7856 to 4.7345). There was only a marginal association between hypercholesterolemia and sensorineural hearing loss (SNHL), with no association between other comorbidities and the type of hearing loss. The current findings raise a need for prioritizing patients with comorbidities in audiological assessment and monitoring in resource-constrained contexts, where capacity versus demand challenges might prevent the provision of audiological services to all adults living with HIV. These findings also highlight the importance of preventive care in this population with regard to the burden of the disease, as it may lead to worse ear and hearing outcomes for affected individuals.

## 1. Introduction

The human immunodeficiency virus (HIV) remains a major public health concern, affecting approximately 40 million people globally [1]. In South Africa, HIV forms part of the challenging quadruple burden of disease [2,3], affecting approximately 8 million individuals (13.5%) [4]. While the introduction of antiretroviral therapy (ART) has resulted in a significant improvement of the immune system in those infected—reducing mortality, improving life expectancy, and improving their quality of life—individuals living with HIV continue to experience various health problems. Sensory pathologies such as hearing loss are among the persistent health-related problems found in adults living with HIV in all stages of the disease, regardless of treatment status [5,6,7,8].

Existing research has demonstrated that adults living with HIV are at an increased risk of hearing loss compared to their HIV-negative counterparts [7,9,10,11,12]. Among the common hearing losses in this population, cochlear hearing loss seems to be the most common auditory pathology [9,13]. While the etiology of cochlear hearing loss is still an ongoing debate in the literature, converging evidence using distortion product otoacoustic emissions (DPOAE) indicates that hearing loss may result from the potential ototoxic nature of highly active antiretroviral therapy (HAART) [8]. Despite this information, there remains a paucity of evidence on the interaction between hearing function and comorbidities in adults living with HIV. The paucity of information on the interaction of these conditions, which may have clinical implications for audiological management, warrants attention, hence the importance of the current study. 

Adults with HIV have been reported to present with comorbidities, over and above the virus [14]. These comorbidities include chronic renal disease [15], hepatitis B and hepatitis C [16], cardiovascular disease [17], dyslipidemia [18], anemia [19], and hypertension, as well as endocrine diseases such as diabetes [15,20]. Gallant and colleagues [14] reported that the prevalence of comorbidities seems to be increasing over time. Lorenc and colleagues [21] reported that 29% of individuals living with HIV have at least one comorbidity. In a study conducted in South Africa by Negin and colleagues [22], 29.6% of adults living with HIV were reported to present with chronic illness.

The comorbid conditions listed above, alone, have also been associated with hearing impairment in adults [23,24]. Hlayisi et al. [25] reported a prevalence of hearing loss of 55% in adults with diabetes compared to 20% in the control group, while Khoza-Shangase et al. [23] found elevated hearing thresholds of 6000 Hz, with elevated speech audiometry findings and reduced DPOAE amplitudes in the high frequencies in this population with diabetes. Yikawe et al. [26] found participants with hypertension to present with a higher prevalence of hearing loss (38.5%) when compared to those without hypertension (13.5%). In a study conducted in South Africa on gold miners exposed to excessive noise, Khoza-Shangase [8] found that participants with a history of tuberculosis (TB) treatment presented with worse hearing thresholds in the high frequencies when compared to those without this history, with clear evidence of a noise-induced hearing loss notch at 6000 Hz in both groups. These findings indicate that comorbidity may increase the likelihood of developing hearing loss in adults; therefore, its co-occurrence with HIV warrants investigation.

Given the reality of a possible co-occurrence of these comorbid conditions with HIV in each individual living with HIV, and given that the prevalence of hearing loss is increasing globally, particularly in low- and middle-income countries (LMICs) due to various risk factors [27], the relevance of the current study within preventive healthcare protocols is raised. The findings from this study have important clinical implications that might allow audiologists and other ear and hearing health professionals to develop appropriate identification and management strategies that are appropriate for adults living with HIV and other underlying comorbidities. This study, as part of a bigger study titled “Wideband Acoustic Immittance in Adults Living with Human Immunodeficiency Virus” had the following objectives:

### Objectives of the Study

To list comorbidities found in a group of adults living with HIV;To describe and compare hearing function in adults living with HIV with comorbidities (study group) and those without comorbidities (comparison group);To determine any association between comorbidities found and hearing function in adults living with HIV.

## 2. Materials and Methods

This study employed a quasi-experimental non-equivalent control group design [28]. This method was chosen and deemed appropriate because participants were not randomly assigned to groups [29], and the researcher did not manipulate any variables [28]. A non-probability purposive sample of 132 adults, with participants who were recruited from one of the HIV clinics in a tertiary hospital in Johannesburg, Gauteng Province, South Africa, comprised the study participants. Participants were recruited and selected if they met the inclusion criteria set for the study. The participants recruited and included in the study were adults aged 18 years and older, diagnosed with HIV, and attending an HIV clinic at the tertiary hospital. Exclusion criteria included adults who presented with otorrhea on the day of testing [30]. The study commenced after ethical approval was secured from the Human Research Ethics Committee (HREC) (Medical) of the University of the Witwatersrand (Protocol number: M190752).

Eligible participants were provided with a participant information sheet that detailed the purpose and nature of the study. All participants provided written informed consent to partake in the study [30]. Following consent, all participants underwent a basic audiological test battery which included case history collection and medical record reviews using a self-administered questionnaire and a data collection form; a video otoscopic evaluation using Firefly Wireless DE550; acoustic immittance testing using Titan 3.3 (Interacoustics, Middelfart, Denmark); and pure tone audiometry testing through the use of the GSI 61 audiometer (Interacoustics, Middelfart, Denmark). Air conduction thresholds were obtained between 0.25 Hz and 8 KHz using Sennheiser HA 200 supra-aural headphones, with the cut-off for normal hearing at ≤25 dBHL. Bone conduction thresholds were also obtained between 0.25 Hz and 8 KHz through the Radio Ear B-70 bone conductor. The air/bone gap criterion used to define the conductive hearing loss component was ≥10 dB [31]. During testing, infection control measures were in place as required for audiological testing [32].

Because this was an exploratory study, no pre-selection of comorbid factors was performed, but these emanated from the record reviews. Therefore, no distinction between the different natures of hearing loss (SNHL, conductive hearing loss (CHL), and mixed hearing loss (MHL)) found in the study was made in the main analysis. However, because of the different pathophysiological processes involved in CHL versus SNHL, a nuanced analysis of comorbid factors found per group (SNHL versus CHL/MHL) was conducted.

### Statistical Analysis

STATA version 15.2 was used to analyze all the data. Both the descriptive and inferential statistics were used to analyze the data for the groups. Categorical variables were summarized using frequencies and percentages, while continuous variables were summarized using median and interquartile range since data were not normally distributed. The normality assumption was assessed using the Shapiro–Wilk test, as well as a histogram plot with a superimposed normal curve. The association between hearing loss and comorbidities was assessed using logistic regression. Both univariate regressions to estimate crude measures of association and adjusted regression to account for the effects of other variables were used. Fisher’s exact analysis was also conducted to determine if there was any association between comorbidities and the type of hearing loss. The results were reported as odds ratios together with the corresponding 95% CI and the associated *p*-value. Bivariate analysis was used to compare the interaction between hearing function and comorbidities in the two groups.

## 3. Results

This study enrolled a total of 132 adults diagnosed with HIV. The demographic characteristics of the participants are summarized in Table 1. The median age was 49 years, with an interquartile range of 41 to 57.5 years; the minimum age was 18 years, while the maximum age was 72 years. The majority of the participants were females (65.2%), with 96.2% of the sample being black. Most participants had acquired secondary education (77.3%). Most of the participants were diagnosed with HIV between the years 2001 and 2005, while most were initiated on ART between 2016 and 2020.

The proportion of the participants who were diagnosed with HIV was higher in 1990–2005, compared to the proportion of ART initiation in the same period. However, the proportion of the participants who were diagnosed with HIV was lower from 2006 to 2020 when compared to the proportion of ART initiation in the same period (Figure 1). The baseline CD4 cell count was recorded in 95.5% of the participants. Most of the patients had a CD4 cell count above 500 cells/uL (43.2%). The median CD4 cell count was 436 cell/uL, with an interquartile range of 266–638 cell/uL.

There was an almost equal split of participants with and without comorbidities, with 49.24% (*n* = 67) participants presenting with comorbidities, while 50.8% (*n* = 65) had none (Table 2). The proportion of those with comorbidities was not significantly different from the proportion of those without comorbidities (*p*-value = 0.7773). Of those with comorbidities, 34.8% had a single underlying condition, 10.6% had two underlying conditions, and 3.8% had three underlying conditions. The most common comorbidities were hypertension (23.5%) and hypercholesterolemia (21.2%). Anemia (4.6%) and asthma (3.8%) were also reported in some of the participants.

### 3.1. Description of Hearing Function

Table 3 provides an overall description of the hearing function outcomes for this study. A total of 30 (22.7%) of the participants presented with hearing loss in this study. Bilateral hearing loss was higher (12.9%) than unilateral hearing loss (9.9%), with the left ear presenting with a higher proportion of hearing loss (6.8%) when compared to the right ear (3.03%). The severity of hearing loss varied across participants. Regarding the left ear, 11.4% had mild hearing loss, 1.5% had moderate hearing loss, and 2.3% had severe hearing loss. Similar for the right ear, 9.9% had mild hearing loss, 3.03% had moderate hearing loss, and 6.8% had severe hearing loss. Therefore, generally, the hearing loss tended to be mild in nature, followed by a severe degree, with no profound degree of hearing loss found. The hearing loss outcomes were also classified according to the type of hearing loss. Regarding the left ear, 5.3% had MHL, and 6.82% had an SNHL abnormality, while 7.6% had CHL; in the right ear, 3.8% had MHL, and 6.1% had SNHL, while 6.1% had CHL. Overall, SNHL was the most prevalent, with MHL being the least occurring.

### 3.2. Hearing Loss and Comorbidities

Table 4 shows the results from the bivariate analysis and logistic regression to determine the association of hearing loss and comorbidities among HIV patients. Among the participants with hearing loss, over half (53.33%) had comorbidities, while 46.67% had no comorbidities. There was no significant difference in these proportions (*p*-value = 0.610). The univariate analysis showed crude odds of 1.236 (95%CI: 0.5467 to 2.795) for the comorbidity covariate. This meant participants with comorbidities were 1.23 times more likely to have hearing loss when compared to those who did not have comorbidities. However, this was not statistically significant (*p*-value = 0.61). The participants with hearing loss were much older (46–61 years), with a median age of 55 years, while those without hearing loss were younger (39–56 years), with a median age of 47.

The more comorbidities the patients had, the higher the odds of developing hearing loss when compared to those without any comorbidity. Those who had three underlying conditions were 2.52 times more likely to develop hearing loss than those without any co-occurring condition. Hypertensive patients were 93% more likely to develop hearing loss when compared to those who were not hypertensive (OR = 1.928; 95%CI: 0.7856 to 4.7345). However, this was not statistically significant (*p*-value = 0.152). Similarly, those patients with hypercholesterolemia were 87% more likely to develop hearing loss when compared to those who had no hypercholesterolemia; this finding was also statistically non-significant (*p*-value = 0.185).

The adjusted regression analysis still did not show any significant association between having comorbidities and having hearing loss in this sample of adults with HIV. However, the estimated effect became protective (AOR = 0.7737; 95%CI: 0.258 to 2.4388). A significant association was observed for the age variable, and a one-year increase in age increased the odds of having hearing loss by 5% (*p*-value = 0.026), adjusting for the other variables. Males had increased odds of 2.2636 (95%CI: 0.91 to 5.64) of having hearing loss, which was marginally significant (*p*-value = 0.079).

With regard to the left ear, most patients with comorbidities had mild hearing loss (64.3%, *n* = 9); however, there was no association between having a comorbidity and the severity of the hearing loss (*p*-value = 0.28). The most common hearing loss was SNHL, and there was a marginal association between having a comorbidity hearing loss type (*p*-value = 0.059). For the right ear, most patients with comorbidities had mild hearing loss (81.82%, *n* = 9); however, there was a marginal association between comorbidity and the severity of the hearing loss (*p*-value = 0.068). The most common hearing loss type was SNHL (*n* = 5, 45.45%), but with no association between participants with a comorbidity and SNHL (*p*-value = 0.861). Fisher’s exact test indicated only a marginal association (*p*-value = 0.664) between hypercholesterolemia and SNHL, with no association between other comorbidities and CHL and/or MHL. 

## 4. Discussion

The aim of the current study was to investigate the interaction between hearing function and comorbidities in adults living with HIV. While research has been conducted to investigate hearing function and HIV, there remains a paucity of evidence on hearing function in adults living with HIV, particularly in those with additional underlying comorbidities. Therefore, the current findings contribute towards this gap in knowledge and lay the groundwork while raising important implications for future research in this population. The current findings on comorbidities found in adults living with HIV are consistent with those previously documented [14], with hypertension and hypercholesterolemia being the most common comorbidities in this population. 

The overall prevalence of hearing loss in this study was 22.73%, and the hearing loss was predominantly bilateral in symmetry (12.88%). The severity and type of hearing loss outcomes varied across different ears, but mild SNHL appeared to be the most common. The existence of hearing loss in adults living with HIV in this study is consistent with the documented literature, indicating that hearing loss is common [6,9,13,33]. Previously published research has attributed hearing loss in adults living with HIV to the virus itself [6], opportunistic infections resulting in middle ear pathologies that are ultimately CHL or MHL [11], and the potential ototoxic nature of the ART regimen [10]. Khoza-Shangase [34] argued that an ototoxicity monitoring protocol for patients who are currently being treated with the ART regimen should be implemented in clinics where HIV is treated to improve early identification of hearing loss in this population. In addition, these monitoring programs may also help identify preventable auditory pathologies, such as middle ear pathologies or hearing loss with a reversible conductive element, through early referral to and treatment by otorhinolaryngologists [34]. 

While hearing loss is generally common in adults living with HIV, the current findings indicate that the occurrence of hearing loss is much higher in participants with comorbidities than those without comorbidities. Although the current results are not statistically significant, participants with comorbidities were 1.23 times more likely to have hearing loss compared to those who did not have comorbidities. The number of comorbidities increased the risk of hearing loss. For example, those who had three underlying comorbid conditions were 2.52 times more likely to present with hearing loss than those without any condition. Among the participants with comorbidities, those with hypertension and hypercholesterolemia were 93% and 87% more likely to present with hearing loss than those who did not have these conditions. These findings are consistent with previous research [35,36]. In research focusing on the interaction between exposure to noise and comorbidities, such as the ones found in adults with HIV, similar findings were revealed, with participants presenting with worse hearing function where comorbidities existed [37].

Although the pathophysiology that explains the increased risk of hearing loss in adults living with HIV and comorbidities is currently unknown, the current authors suspect that the combined effects of comorbid conditions and HIV may play a role. Available evidence suggests that hypertension and hypercholesterolemia compromise the vascular supply which may result in hearing loss [36], and HIV increases the risk of hearing loss because of the ART regimen and opportunistic infections [10,11]. Therefore, future studies using case–control research methodologies are needed to investigate this pathophysiology further.

Further analysis of the results revealed that mild hearing loss was common in this sample. However, there was no statistically significant association between severity and having comorbidities (*p* = 0.028). Although CHL and SNHL were both common, there was no statistically significant association between comorbidities and the type of hearing loss found. Further analysis indicated that there was only a marginal association between hypercholesterolemia and SNHL, but there was no association between other comorbidities and other types of hearing loss. This means that patients with comorbidities can present with hearing loss of any type and severity. Despite the severity and/or type, hearing loss can have a significant impact on the individual’s quality of life. Roup [38] reported that adults with mild SNHL can have significant difficulties in the perception of speech in the presence of background noise. Davies and colleagues [39] demonstrated that adults with varying degrees of hearing loss experience listening-related fatigue, which may significantly compromise their quality of life [40]. Future studies are needed to determine the impact of the severity/type of hearing loss in adults living with HIV with underlying comorbidities.

### Study Limitations

Although this study provides new evidence on hearing function in adults living with HIV while having underlying comorbidities, there are methodological limitations that need to be taken into account during the interpretation of the findings. First, the sub-sample of participants with hearing loss was small, limiting the generalizability of the findings. This raises implications for future collaborative studies from different sites exploring the same questions. Secondly, hearing function in this study was primarily based on standard pure tone audiometry. However, Khoza-Shangase [10] demonstrated that hearing impairment in adults living with HIV may occur in the ultra-high frequencies, occurring before it can be detected by standard pure tone audiometry. Future studies need to include measures such as otoacoustic emissions and ultra-high frequency audiometry to identify early signs of hearing impairment. Lastly, no age or gender matching of participants in the study versus the comparison group was conducted; therefore, the influence of these variables on the current findings could not be controlled for, which is another implication for future studies.

## 5. Conclusions

The current findings indicate that the prevalence of hearing loss in adults living with HIV and comorbidities is higher than in those without comorbidities. The higher the number of comorbidities, the more likely the patients will develop hearing loss. The participants with hypertension and hypercholesterolemia were more likely to develop hearing loss than those without these conditions or with other comorbid conditions. The participants with hearing loss were also found to be older than those without hearing loss. These findings raise important implications for preventive healthcare in general, over and above preventive audiology, in particular. Some of the comorbidities are diseases of lifestyle that fall under the quadruple burden of disease that South Africa is grappling with. Preventive efforts directed at minimizing and/or eliminating these comorbidities will not only improve the health and quality of life of the individuals involved but will also serve as a preventive measure for hearing loss in this population [41]. Given that the prevalence of HIV is high in South Africa and other LMICs, and demand versus capacity challenges continue to exist [33], the current findings raise important preventive healthcare and preventive audiology implications. Audiologists must engage in all levels of preventive care from the primordial to the tertiary level, with innovative service delivery models such as the use of trained non-professionals in task shifting, and tele-audiology models of service delivery, with the goal to implement successful sustainable preventive audiology programs within the South African population of adults living with HIV.

## Figures and Tables

**Figure 1 ijerph-18-12177-f001:**
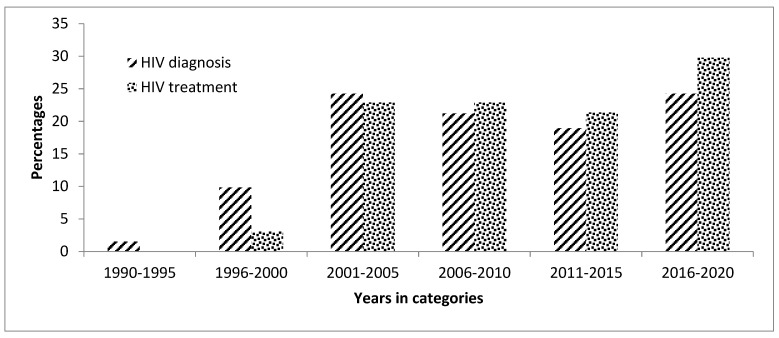
Distribution of antiretroviral therapy (ART) diagnosis and ART initiation in the study participants (*n* = 132).

**Table 1 ijerph-18-12177-t001:** Baseline characteristics of the participants (*n* = 132).

Variable	Categories	Frequencies	Percentages
Gender	Female Male	86 46	65.15 34.85
Ethnicity	Black White Colored	127 1 4	96.21 0.76 3.03
Level of Education	Primary Secondary Tertiary	20 102 10	15.15 77.27 7.58
Year of HIV Diagnosis	1990–1995 1996–2000 2001–2005 2006–2010 2011–2015 2016–2020	2 13 32 28 25 32	1.52 9.89 24.24 21.21 18.94 24.24
Year Patient Initiated on ART	1995–2000 2001–2005 2006–2010 2011–2015 2016–2020	4 30 30 28 39	3.05 22.9 22.9 21.37 29.77
CD4 Categories	<200 201–350 351–500 >500 Missing	18 28 23 57 6	13.64 21.21 17.42 43.18 4.55

**Table 2 ijerph-18-12177-t002:** Frequency of comorbidities in the adults with HIV (*n* = 132).

Variable	Categories	Frequencies	Percentages
Has comorbidities	No Yes	67 65	50.76 49.24
Number of comorbidities	0 1 2 3	67 46 14 5	50.76 34.85 10.61 3.79
Types of comorbidities (yes category only)	Hypertension Hypercholesterolemia Anemia Asthma Atopic dermatitis Chronic hepatitis B Cryptococcosis Diabetes mellitus Lipodystrophy Mixed hyperlipidemia Peptic ulcer Arthritis Sinusitis GORD Myalgia Polyneuropathy Rhinitis	31 28 6 5 1 3 2 3 1 1 1 1 1 1 1 1 1	23.48 21.21 4.55 3.79 0.76 2.27 1.52 2.2 0.76 0.76 0.76 0.76 0.76 0.76 0.76 0.76 0.76

**Table 3 ijerph-18-12177-t003:** Hearing function outcomes in the sample (*n* = 132).

Variable	Categories	Frequencies	Percentages
Overall hearing loss	No Yes	102 30	77.27 22.73
Laterality	No Unilateral Left Unilateral Right Bilateral	102 9 4 17	77.27 6.82 3.03 12.88
Left severity	No left ear problem Mild Moderate Severe	112 15 2 3	84.85 11.36 1.52 2.27
Right severity	No right ear problem Mild Moderate Severe	106 13 4 9	80.3 9.85 3.03 6.82
Left ear abnormality patterns	Normal Mixed Sensory Conductive	106 7 9 10	80.30 5.30 6.82 7.58
Right ear abnormality patterns	Normal Mixed Sensory Conductive	111 5 8 8	84.09 3.79 6.06 6.06

**Table 4 ijerph-18-12177-t004:** Logistic regression of hearing outcomes and risk factors adjusting for the effects of other variables.

Variable	Bivariate Analysis			Univariate		Multiple Regression	
	HF * No *n* (%)	HF * Yes *n* (%)	*p*-Value	OR (95%CI)	*p*-Value	AOR (95%CI)	*p*-Value
Comorbidities							
No	53 (51.96)	14 (46.67)		reference		reference	
Yes	49 (48.04)	16 (53.33)	0.610	1.236 (0.5467 to 2.795)	0.610	0.7937 (0.2583 to 2.4388)	0.687
Age							
Median (IQR)	47 (39–56)	55 (46–61)	0.0031	1.064 (1.019 to 1.109)	0.004	1.0539 (1.006 to 1.1036)	0.026
Sex							
Female	70 (68.63)	16 (53.33)		reference		reference	
Male	32 (31.37)	14 (46.67)	0.122	1.914 (0.834 to 4.391)	0.125	2.2636 (0.9089 to 5.6379)	0.079
Education							
Primary	12 (11.76)	8 (26.67)		reference		reference	
Secondary	81 (79.41)	21 (70.0)		0.3889 (0.141 to 1.073)	0.068	0.4603 (0.1552 to 1.3651)	0.162
Tertiary	9 (8.82)	1 (3.33)	0.101	0.1667 (0.018 to 1.583)	0.119	0.2081 (0.0202 to 2.1432)	0.187
CD4 cell count							
Median (IQR)	423 (262–641)	519 (352–638)	0.1403	1.001 (0.999 to 1.0003)	0.063		
Number of comorbidities							
0	53 (51.96)	14 (46.67)		reference			
1	36 (35.29)	10 (33.33)	0.735	1.051 (0.4211 to 2.6268)	0.914		
2	10 (9.8)	4 (13.33)		1.5143 (0.4125 to 5.5593)	0.532		
3	3 (2.94)	2 (6.67)		2.5238 (0.3837 to 16.6001)	0.335		
Hypertension							
No	81 (79.41)	20 (66.67)		reference		reference	
Yes	21 (20.59)	10 (33.33)	0.148	1.928 (0.7856 to 4.7345)	0.152	1.5141 (0.4335 to 5.2875)	0.516
Hyper-cholesterolemia							
No	83 (81.37)	21 (70.0)		reference			
Yes	19 (18.63)	9 (30.0)	0.180	1.8722 (0.7413 to 4.7281)	0.185		

HF * = hearing function.

## Data Availability

Data are available from the corresponding author, B.S., upon reasonable request.

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
