# Peer review of "Investigation of the Interaction between Hearing Function and Comorbidities in Adults Living with Human Immunodeficiency Virus"

_ijerph, 2021, doi:10.3390/ijerph182212177_

Round 1

Reviewer 1 Report

The objective of this study was to determine and compare the hearing function of a group of adults living with HIV and comorbidities and those without comorbidities.

Hearing impairment has been documented as one the manifestations in HIV-seropositive individuals. Sensorineural hearing loss has been attributed to ototoxicity of antiretroviral drugs or a direct effect of HIV on the CNS, peripheral nerves, and cochlea-vestibular apparatus or due to secondary opportunistic infections or toxicity of drugs to treat these infections. Some studies showed that there was no correlation between CD4 T-lymphocyte count and hearing loss. Others found abnormal hearing impairment in subjects with prolonged HIV infection, viral load, and low CD4. However, it seems that hear impairment had no relation with ART, duration of treatment, or severity of the disease, indicating that cochlear functions are impaired by HIV either directly or indirectly through premature aging, similar to accelerate aging of the auditory system. Increasing evidence now suggests that HIV individuals experience accelerated changes on the cochlea or central auditory system similar to those observed in elderly persons with accelerate aging and related comorbidities.

So, the evaluation of hearing function among all patients diagnoses with HIV seems to be an accepted approach.

Two recent published studies (de Jong MA et al. Front Neurol. 2019;10:845; Minhas RS et al. Int Arch Otorhinolaryngol. 2018;22(4):378-381).

Underlined the increased risk of hearing loss among HIV infected persons and recommended early detection and the use of several auditory measurements.

The conclusions of this study are on line with those recently published studies, however seems now clear that the risks of hearing loss experienced in HIV individuals are similar to the risks observed in elderly persons with accelerate aging and related comorbidities, and not with ART, duration of treatment or severity of the disease. This study would be enriched with stratification of ages (e.g. less or more 35 years), and matching with HIV-negative controls.

The conclusion of the study indicating that the prevalence of hearing loss in adults living with HIV and having comorbidities was higher than in those without comorbidities it’s not surprising taking in account two studies mentioned above. The same to the high the number of comorbidities, participants with hypertension and hypercholesterolemia that were more likely to develop hearing loss than those without these conditions.

As it was mentioned by the authors the results of this study can be useful to implement successful sustainable preventive audiology programmes within South African populations of adults living with HIV, but do not add much more to the published studies.

Others comments:

  • Some numbers of references in the text do not match with the numbers of listed references (e.g. 15, 21, 22, 23, 14,…)
  • References – should be reviewed carefully (e.g. 13 – year missed; 22 – name of the article; 27 and 34 – names of the journals; 37 – names of the authors).

Author Response

Reviewer’s comment

Response and action

The conclusions of this study are on line with those recently published studies, however seems now clear that the risks of hearing loss experienced in HIV individuals are similar to the risks observed in elderly persons with accelerate aging and related comorbidities, and not with ART, duration of treatment or severity of the disease.

This study would be enriched with stratification of ages (e.g. less or more 35 years), and matching with HIV-negative controls.

Due to the lack of control group, this matching could not be done. However, we detailed this as a limitation of the study and the need for future research.

Stratification of age was included (line 176-177), and also under conclusion (line 283-284)

Some numbers of references in the text do not match with the numbers of listed references (e.g. 15, 21, 22, 23, 14,…)

Addressed

References – should be reviewed carefully (e.g. 13 – year missed; 22 – name of the article; 27 and 34 – names of the journals; 37 – names of the author

Addressed

Reviewer 2 Report

The study is well designed.

My principal observation is about the overall consideration of subjects affected by sensory-neural hearing loss and mixed-conductive hearing loss.

These are two different populations in which the pathological effects of virus are different, in the first case the effect is neurotoxic while in the second one is mediated through a middle ear inflammation (acute or chronic otitis media), but even a casual correlation with hearing loss may be  hypothesized (otosclerosis). 

Since comorbidities considered comprise pathologies that affect inner ear I my suggestion is that the Author must re-evaluate data excluding by the sample patients affected by mixed-conductive hearing loss. 

Author Response

Reviewer’s comment

Action

My principal observation is about the overall consideration of subjects affected by sensory-neural hearing loss and mixed-conductive hearing loss.

These are two different populations in which the pathological effects of virus are different, in the first case the effect is neurotoxic while in the second one is mediated through a middle ear inflammation (acute or chronic otitis media), but even a casual correlation with hearing loss may be  hypothesized (otosclerosis). 

Since comorbidities considered comprise pathologies that affect inner ear I my suggestion is that the Author must re-evaluate data excluding by the sample patients affected by mixed-conductive hearing loss. 

  1. SNHL in HIV is not only neurotoxicity linked, but ototoxicity linked as well, with primary effects also documented (primary is where the virus itself causes hearing loss – because its neurotropic in nature)
  2. The comorbidities were not pre-selected, they emanated from the data

However, we re-evaluated the data as per the reviewer’s comment and we found that there was only a marginal association between one comorbidity (hypercholesterolemia) and SNHL. There was no association found between other comorbidities and CHL/MHL.

This sub-analysis and reporting was included in the following sections:

Abstract: line 19-20

Methods section: line 113-118

Statistical analysis: 128-129

Result section: line 214-216

Discussion section: 266-268

Round 2

Reviewer 1 Report

Abbreviations should be defined the first time they appear in each of three sections: the abstract; the main text; the first figure or text.

Abstract

Line 9 – correct to “…human immunodeficiency virus (HIV)…”

Line 19 – correct to “…sensorineural hearing loss (SNHL)…”

Introduction

Line 52 – correct to “…dyslipidemia (18), anemia (19) …”

Line 115 – correct to “…[SNHL, conductive hearing loss (CHL), and mixed hearing loss (MHL)]…”

Line 150 – correct to “Figure 1. Distribution of antiretroviral therapy (ART) diagnosis…”

Line 171 – correct to “…5,3% had a MHL,…”

Line 172 – correct to “…had a CHL…”

Table 4 – Abbreviation HRyes may be HFyes and line up HFno with HFyes and use the asterisk HF*

Line 233-234 – correct to “…ultimately CHL or MHL (11)…”

References

Abbreviate all Journal names

Names of articles – standardize (do not use a capital letter to the first letter of each word, except the cases of own names)

Reviewer 2 Report

After the revision the main problem of the paper has not been resolved since the study represents an analysis of audiometric results in these group of patients without any evaluation about the kind of hearing deficit and about the different effect that HIV can have on the middle or inner ear function.
